# Knowledge and Attitudes towards Patient Safety among Students in Physical Therapy in Spain: A Longitudinal Study

**DOI:** 10.3390/ijerph191811618

**Published:** 2022-09-15

**Authors:** Joaquina Montilla-Herrador, José A. Lozano-Meca, Aitor Baño-Alcaraz, Carmen Lillo-Navarro, Rodrigo Martín-San Agustín, Mariano Gacto-Sánchez

**Affiliations:** 1Department of Physical Therapy, Faculty of Medicine, CEIR Campus Mare Nostrum (CMN), University of Murcia, 30100 Murcia, Spain; 2Instituto Murciano de Investigación Biosanitaria-Virgen de la Arrixaca (IMIB-Arrixaca), El Palmar, 30120 Murcia, Spain; 3Department of Pathology and Surgery and Center for Translational Research in Physical Therapy (CEIT), University Miguel Hernández, Sant Joan, 03202 Alicante, Spain; 4Department of Physical Therapy, University of Valencia, 46010 Valencia, Spain

**Keywords:** health quality management, physical therapy, patient safety, attitudes

## Abstract

(1) Background: Patient safety is a discipline of health care management aiming to prevent and reduce errors and harm to patients. The assessment of knowledge and attitudes on patient safety among students in physical therapy is still scarce; no studies have yet explored the changes that internship periods may produce. Objectives: 1. to determine the attitudes and knowledge of students in physical therapy with respect to patient safety in a Spanish University; and 2. to explore changes following a practical internship period. (2) Methods: Longitudinal study. Data from the Attitudes to Patient Safety Questionnaire III (APSQ-III) before and after the internship period were obtained from an initial sample of 125 students and average positive response rates were compared. (3) Results: “Team functioning”, “Importance of patient safety in the curriculum”, and “Error inevitability” displayed the highest scores, in accordance with the current literature. After the internship period, the dimensions “Patient safety training received” (*p* = 0.001), “Error reporting confidence” (*p* = 0.044), and “Professional incompetence as an error cause” (*p* = 0.027) showed significant changes. (4) Conclusions: The current study, highlighting areas of strengths and weaknesses in the knowledge and attitudes of students in physical therapy towards patient safety, may be a foundation to adopt tailored programs to enhance students’ competencies in patient safety.

## 1. Introduction

Patient Safety is a discipline of Health Care Management aiming to prevent and reduce risks, errors, and harm that befall patients during the provision of health care [1]. Avoidable patient harm is therefore a major public health concern [2], since the occurrence of adverse events related to unsafe health care is one of the ten leading causes of death and/or disability worldwide [3]; furthermore, around one in every ten patients is harmed while receiving hospital care in high-income countries [4] whilst low and middle-income countries display higher figures, resulting in 2.6 million deaths per year [5].

A safe and patient-centered health approach is closely related to the quality of the training undergone by healthcare students: the implementation of educational and formative programs within the curriculum strengthens the knowledge of patient safety and, subsequently, enhances the development of error prevention strategies [6]. Fostering a “safety culture” within health organizations and universities is identified as a cornerstone and key strategy to guarantee adequate levels of patient safety, alongside reducing medical errors and improving health care delivery [7].

Attitudes towards patient safety correspond to the beliefs and values that professionals have on the perception of security in their workplace. Students and professionals in their first steps usually show more positive attitudes regarding patient security in areas such as teamwork, error inevitability, and the presence of specific curricular training on patient security [8,9].

Several instruments to assess the knowledge and attitudes towards patient safety currently exist [10,11]. In this framework, the “Attitudes to Patient Safety Questionnaire” (APSQ-III, hereinafter) is a widely used instrument with outstanding psychometric properties [12] and has been validated in Spanish in a version with 7 response options [13] and with 5 response options [14].

The current knowledge and attitudes towards patient safety have already been explored by students in the fields of nursing and medicine [15,16,17,18,19,20], globally concluding that the implementation of specific curricular educational programs on patient safety enhances knowledge and attitudes. Wetzel et al. [19] state that, despite the fact that most students hold an adequate level of attitudes about patient safety, both the causes of error and the effects of long work hours on safety require greater curricular attention, alongside the need to improve skills and confidence in error reporting and disclosure. Results from the study by Nadarajan et al. [9] highlight the need to enhance perceptions of disclosure responsibility, professional incompetence, and safety curriculum among medical students in Malaysia.

The assessment of knowledge and attitudes across students in physical therapy is still scarce: Struessel et al. [21] recently developed a tailored program consisting of nine sessions of systems-focused patient safety curricular content. Changes in knowledge and attitudes towards patient safety were assessed via a modified version of the APSQ-III. Results stemming from the aforementioned study show that statistically significant changes in the mean response rates involved four of the nine dimensions of the APSQ-III.

Practical internship periods are usually clinical periods for putting knowledge into practice in order to complete a specific course or subject. As a professional immersion period, internship periods create an excellent environment to enhance perceptions, professional confidence, and competences [22,23]. Supervised practice experiences are of great importance for students to internalize and apply the principles of patient safety in clinical practice [24]. In the field of physical therapy, results show that internship programs have a positive impact in terms of clinical practice, confidence, employment opportunities, productivity, and government policy, besides being an excellent approach to reduce the existing theory-practice gap [25,26]. To date, no studies have yet explored the changes that internship periods may have on the knowledge and attitudes towards patient safety for students in physical therapy.

The objective of the current study was, therefore, (1) to determine the levels of attitudes and knowledge of students in physical therapy with respect to patient safety in a Spanish University and (2) to assess changes in students’ knowledge and attitudes across a practical internship period.

## 2. Materials and Methods

A longitudinal study was conducted in the degree program for physical therapy at the University of Murcia, Murcia, Spain from October 2021 to June 2022. The study was approved by the University of Murcia Ethics Committee under the code 3577/2021. There were approximately 132 students capable of undergoing a practical internship period in the degree, therefore including students from the 2nd semester of the 3rd year of studies and the 1st semester of the 4th year of studies. Students from fourth 4th year had received neither specific nor general content on PS with respect to their peers from 3rd year. Assuming a proportion for positive perception of 50%, with a confidence level of 95% and setting alpha at 5%, the minimum required sample size was 82 subjects [27].

The study was initially presented to the students in class, highlighting the voluntary character of the participation. Respondents’ written informed consent was collected prior to the study onset. Whenever the student was unable to answer on paper and/or was unavailable for the second (post-test) measure, a tailored Google Form questionnaire was sent to his/her university mail address.

The survey consisted of the adaptation of the 26-item Attitudes to Patient Safety Questionnaire (APSQ-III) [12], given to students in physical therapy, as developed by Struessel et al. [21]: questions from the APSQ-III were slightly adapted to apply to physical therapist student education and physical therapy practice. For instance, the term “doctor” or “clinician” was changed into “physical therapist” and “medical school training” was changed to “physical therapy education”. Two specific items from APSQ-III (focusing on careless nurses and doctors) were collapsed into a single question concerning careless “healthcare providers” [21]. Three of the authors (J.M.-H., J.A.L.-M. and M.G.-S.) determined that the new version adequately covers the concept measured, therefore confirming the face validity of the tool, defined as the “extent to which a test is subjectively viewed as covering the concept it purports to measure” [28].

We adapted the response options, from the initial approach (i.e., a 1-to-7 Likert scale) to the widely used approach consisting of a Likert scale in a 1-to-5 range, as stated elsewhere [9,29,30], in which “1” corresponds to total disagreement and “5” stands for total agreement with the statement expounded in the corresponding item (Appendix A). The scoring system was based on the criteria defined by Wetzel et al. [20] and adopted in previous research [9]: the desired response was defined as “strongly agree” or “agree” except for the reverse-coded items (items 11, 13–17, and 24 from our 25-item adapted scale), for which “disagree” or “strongly disagree” was the desired response. Responses were subsequently dichotomized into 1 (positive) or 0 (negative).

The 26-item questionnaire encompasses a total of nine safety domains:(1)Patient safety training received (ST);(2)Error reporting confidence (ER);(3)Working hours as an error cause (WH);(4)Error inevitability (EI);(5)Professional incompetence as an error cause (PR);(6)Disclosure responsibility (DR);(7)Team functioning (TF);(8)Patient involvement in reducing error (PA);(9)Importance of patient safety in the curriculum (SC).

The average positive response rate (APRR, hereinafter) for each APSQ-III domain was then calculated as the average percentage of the positive perception statements [19]. In order to decide the desirable level of positive attitude for each domain, the cut-off point of 75%, known as the Nordén–Häag criterion, was adopted [30].

Students were assessed regarding the APSQ-III questionnaire at baseline, in a first session, and then after the internship period. The internship period took place in different settings (hospitals, primary healthcare centers, private clinics, centers for accident insurance companies, patients’ associations, or geriatric wards): the activities and competences planned by the University did not differ between 3rd and 4th year students. Tutors adapt the activities to their own clinical daily requirements, to their patients’ characteristics, to the knowledge and skills that they specifically detect as potentially improvable in each student, and to their own teaching style. For students in 3rd year, the internship period covers 5 weeks (at the end of the 2nd semester), whereas an 8-week period is organized for 4th year students (during their 1st semester). The differential subjects between 3rd and 4th year of studies (i.e., the subjects already studied by 4th year students but yet to be taught to 3rd year students) have no specific content regarding patient safety, beyond the general recommendations that lecturers may provide in their regular teaching activity (i.e., the “hidden curriculum” [24]).

The percentage of positive responses was compared, across dimensions, between the pre- and post-test measures (before and after undergoing the corresponding practical internship period). Data were tested for normality through the Kolmogorov–Smirnov test and, subsequently, parametric or non-parametric contrasts were used for this purpose.

All analyses assumed a two-sided test of hypothesis and *p*-values < 0.05 were considered statistically significant. The statistical package SPSS (IBM SPSS Statistics, version 23.0, IBM Corporation, Armonk, NY, USA) was used for all the analyses.

## 3. Results

A total of 125 students were recruited. Participants’ baseline characteristics are presented in Table 1: gender was balanced in the sample (66 subjects were female; 52.8%). Mean age was 22.04 ± 4.12 years. Roughly half of the students were in 4th year. Solely 4% (5 students) were working in health-related positions at the time of the survey. Most of the students were admitted to the University from high school studies (104 subjects; 83.2%). A high percentage of the sample had previously encountered no specific content regarding patient security (70 students; 56.0%): although no specific content regarding patient safety is directly included in the curriculum, some students had previous experiences and or acquired competences based, for instance, on previous educational and/or academic experiences (as, for instance, in training and development degrees previously studied), or courses or formal education studied besides the degree in physical therapy. A total of 100 students had previous experience in physical therapy services as patients (80%) at the time of the initial assessment.

Mean scores of the different dimensions are presented in Table 2. For the pre-test assessment, straight scores ranged from 7.426 ± 2.096 to 11.363 ± 2.469, values corresponding to the dimensions “Professional incompetence as an error cause” and “Working hours as an error cause”, respectively. Concerning the post-test measure, “Patient involvement in reducing error” showed the lowest mean value (8.009 ± 1.597) whilst “Working hours as an error cause” displayed the highest mean (12.647± 2.556). Mean responders (across the nine dimensions) to the first assessment were 121.44 subjects, whereas the mean for the nine dimensions of the second measure was 104.11 individuals, therefore entailing a total drop-out rate of 14.27%. Table 2 also displays the percentage of positive responses or “Average Positive Response Rates” (APRR) according to the criteria defined by Nordén–Häag [30], with scores in a 44.262 to 91.803 percentage range (“Error reporting” and “Team functioning”, respectively) in the first measure, and from 42.539% to 87.142% (“Professional incompetence as an error cause” and “Team functioning”, respectively) in the second assessment. No statistically significant differences per gender were stated.

Data for APRR were not normally distributed (Kolmogorov–Smirnov test; *p*-value = 0.017), therefore justifying the use of non-parametric approaches to compare the changes in APRR between the first and second assessments. Since mean percentages were paired (student response at baseline; same student response after the practical internship period), the Wilcoxon signed-rank test was used to compare the results.

Table 3 displays the results of the Wilcoxon signed-rank test. The changes in the dimensions “Patient safety training received” (*p* = 0.001), “Error reporting confidence” (*p* = 0.044), and “Professional incompetence as an error cause” (*p* = 0.027) showed statistically significant differences. Differences observed across the extant six dimensions of the APSQ-III did not reach statistical significance, although the dimensions “Working hours as an error cause” (*p* = 0.058) and “Importance of patient safety in the curriculum” (*p* = 0.061) showed differences bordering on statistical significance.

Further visual information on the percentage rates, changes, and statistical significances is provided in Figure 1.

## 4. Discussion

The current study had both a descriptive and exploratory target, since the objectives were to determine the levels of attitudes and knowledge of students in physical therapy in Spain with respect to patient safety, and to explore and assess changes in students’ knowledge and attitudes following a practical internship period.

The first assessment performed displayed the three following dimensions as those with the highest APRRs: “Team functioning” (91.803%), “Importance of patient safety in the curriculum” (85.792%), and “Error inevitability” (84.972%). These findings are in line with the results from a recent systematic review, in which the fact that healthcare students and young professionals showed overwhelmingly positive patient safety attitudes in some areas such as teamwork, climate, error inevitability, or importance of patient safety in the curriculum is highlighted [8].

The initial measurement performed on the different dimensions of the APSQ-III expounds average positive response rates below those reported by other studies across seven dimensions: APRR for “Patient safety training received” reached approximately 56%, compared to 85% reported by Nadarajan et al. [9], 67% stated by Struessel et al. [21], and 83.5% reported by Wetzel et al. [19]. The low score may be influenced by the fact that patient safety tends to be considered implicit in the curricula as an overall program outcome, rather than a distinct competency area [24].

Concerning the dimension “Error reporting confidence”, our scores reveal a percentage of 44%, widely below the results from Nadarajan, Struessel, and Wetzel (76.3%, 75%, and 57.6%, respectively). Students have disclosed that they felt uncomfortable towards the idea of reporting errors: the results of our study may be influenced by the uncertainty of the students, since they were about to start their clinical placement in a center with an unfamiliar professional tutor in a few days’ time. On another note, Creswell et al. highlight the fact that teaching is predominantly based on idealized settings, which offer limited opportunities for interprofessional learning. As a result, for example, there are usually few opportunities for students to learn about organizational policies and procedures, such as incident reporting [24].

On “Working hours as an error cause” our results, bordering a 70% of APRR, are still below the rates from other authors (Nadarajan reports 89.5%, Struessel states 82%, whilst Wetzel expounds 73.4%). Struessel et al. [21] entitle this dimension as “Situational awareness”, unlike the original appellation defined by Carruthers et al. [12].

Focusing on “Professional incompetence as an error cause”, our rates (roughly 51%) are much lower than those from Nadarajan (70%), Struessel (65%), and Wetzel (64%). “Disclosure responsibility” displayed, in our study, an APRR close to 50% versus 68.5% (Nadarajan), 79% (Struessel), and 65.7% (Wetzel). “Patient involvement in reducing error” also scored below the aforementioned authors: 70% in our study versus 80%, 82%, and 88%, respectively, for Nadarajan, Struessel, and Wetzel. “Error inevitability” APRR corresponded to 85%, compared to 86.1% (Nadarajan), 97% (Struessel), and 92.8% (Wetzel).

Solely two dimensions (“Team functioning” (TF) and “Importance of patient safety in the curriculum” (SC)) were somehow in line with the rates reported elsewhere, since our figures corresponded to 91.8% for TF against 94.6% (Nadarajan), 93% (Struessel), and 88.8% (Wetzel); our rates in SC corresponded to almost 86% versus 80.1% (Nadarajan), 98% (Struessel), and 80.1% (Wetzel).

Thus, overall, the average positive response rates stemming from our study are lower than those reported by other authors: a feasible explanation could lie in the characteristics of the sample itself. Nadarajan and Wetzel developed their respective studies on students in medicine: the curricular design, academic and further professional competencies widely differ with respect to physical therapy. A high percentage (59.7%) of the sample from the study implemented by Wetzel had prior healthcare experience to a certain extent (as physician-shadowing or health-related volunteer working), against an exiguous percentage of 4% of students from our study having experience in health-related professional positions. Furthermore, Wetzel reports that 23% of students indicated some experience as paid, short-term positions whilst 9.3% reported a previous long-term, paid position or prior career in health care. The student sample from Wetzel therefore seems to be more experienced from a professional healthcare perspective, aside from the fact that it stems from the fourth academic year of medicine (our study consisted of a blended sample of third and/or fourth year students). The study from Struessel et al. was performed across a sample of students in physical therapy: students had already completed two academic semesters and two 1-week clinical education experiences. Despite the fact that the initial survey was developed prior to the delivery of any systems-focused patient safety content, the level of studies corresponded to a Doctor of Physical Therapy (DPT) program: typically, DPT programs require students to have earned a bachelor’s degree before entry. A more robust previous background in general healthcare is therefore plausible with respect to our sample. This fact is endorsed by the mean class age upon matriculation (24 years of age) from their study (against the approximate mean age of 22 years in our sample), despite the fact that authors did not collect demographics beyond the overall class demographics to ensure anonymity in the small sample size.

Potential comparisons with other studies having used the APSQ framework are unsuitable, since some other authors used the APSQ-IV instead of the APSQ-III assessment tool [17], calculated their results from the raw scores of the different items and/or dimensions [14] instead of transforming scores into average positive response rates based on Nordén–Häag criteria, or adopted the aforementioned criteria but did not consider items with reverse scores, so that responses were all systematically grouped into agree (4 or 5) and disagree (1 or 2) to obtain the overall percentages [20], a fact that hinders any possible comparison through homogeneous scoring methods.

Focusing on the second objective, to the best of our knowledge, this is the first study to delve into the changes experienced in physical therapy students’ attitudes and knowledge towards patient safety following a clinical internship period. Dimensions that experienced statistically significant changes in our study corresponded to “Patient safety training received”, “Error reporting confidence”, and “Professional incompetence as an error cause”.

Struessel et al., in their previous study focusing on changes following a longitudinal system-focused patient safety curriculum (nine sessions of patient safety and interprofessional curricular content) plus 22 weeks of clinical education, found statistically significant changes for four of the nine APSQ-III dimensions: “Patient safety training received”, “Working hours as an error cause”, “Professional incompetence as an error cause”, and “Disclosure responsibility” [21]. To wit, two dimensions coincide (“Patient safety training received”, and “Professional incompetence as an error cause”): despite this fact, both educational programs and samples widely differ. Currently, the scores from each dimension on patient safety, before and after the practical internship period, might represent positive feedback for stakeholders to integrate patient safety education into campus-based activities and healthcare settings. Future research should consider these findings in order to establish further conclusions in this area.

The results from our study should be interpreted in light of its methodological limitations. First, as in all surveys using self-reported questionnaires, self-report bias remains possible, even though the assessors stressed the voluntary character of the survey. Second, some responders were lost from the initial to the final assessment; despite this fact, the drop-out rate achieved was lower than 15%, therefore indicating low risk for attrition bias [31]. Third, the questionnaire used (APSQ-III) has shown good psychometric properties (e.g., good stability of factor structure, good criterion validity) but some other psychometric aspects (such as the predictive validity of the measure) are yet to be assessed [12]. Fourth, the influence of the physical therapy-related field of the internship, the type of setting, or the tutor could have a fundamental impact on the attitudes and knowledge regarding patient safety in students, and further studies in the field should therefore explore and delve into these potential influences. Finally, the lack of research on the effects of a clinical internship period on the attitudes and knowledge towards patient safety in physical therapy students represents an important knowledge gap in which the current study is framed but, on another note, this scarcity of contextual research hinders potential comparisons and contrasts. Further research should therefore be conducted with the aim of developing thorough conclusions in this research line.

## 5. Conclusions

“Team functioning”, “Importance of patient safety in the curriculum”, and “Error inevitability” displayed the highest scores for average positive response rates, in accordance with the current research literature. Following the clinical internship period, the dimensions “Patient safety training received”, “Error reporting confidence”, and “Professional incompetence as an error cause” showed significant changes. The current study highlights areas of strength and weaknesses in the knowledge and attitudes of students in physical therapy regarding patient safety. This study might be a foundation to adopt tailored programs to enhance students’ competencies in patient safety.

## Figures and Tables

**Figure 1 ijerph-19-11618-f001:**
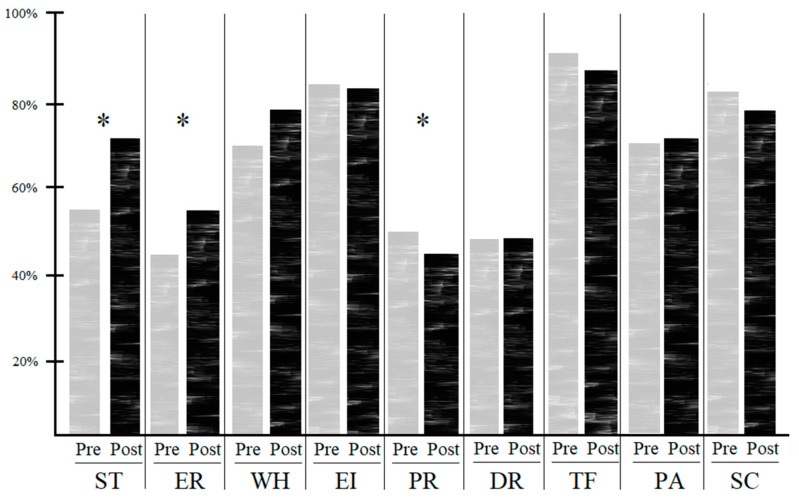
Comparison of average positive response rates per dimension of the APSQ-III (pre vs. post). Abbreviations: ST: Patient safety training received; ER: error reporting confidence; WH: working hours as an error cause; EI: error inevitability; PR: professional incompetence as an error cause; DR: disclosure responsibility; TF: team functioning; PA: patient involvement in reducing error; SC: importance of patient safety in the curriculum; * Significant differences (*p*-value < 0.05) stated through the Wilcoxon signed-rank test.

**Table 1 ijerph-19-11618-t001:** Participants’ characteristics at baseline (*n* = 125).

Variables		Mean ± SD or Frequency (%)
Age		22.04 ± 4.12
Gender	Male	59 (47.2)
	Female	66 (52.8)
Academic Year	3rd	54 (43.2)
	4th	64 (51.2)
	Both	7 (5.6)
Current Professional Position	Not working	97 (77.6)
	Working in a health non-related area	23 (18.4)
	Working in health area	5 (4.0)
Previous Studies	High School	104 (83.2)
	University	4 (3.2)
	Training & Development	6 (4.8)
	Training & Development + High School	11 (8.8)
Encountered Previous Content in Patient Security	No	70 (56.0)
	Yes	55 (44.0)
Previous Experiences in Physical Therapy (as a patient)	No	25 (20.0)
	Yes	100 (80.0)
Number of Visits to Physical Therapy (as a patient)	<10 visits	64 (51.2)
	10–20 visits	47 (37.6)
	>20 visits	14 (11.2)

**Table 2 ijerph-19-11618-t002:** Descriptive statistics of APSQ-III dimensions and average positive response rate per APSQ-III dimension at pre- and post-test.

Dimensions	N	Straight Scores		Scores with Reversed Items		Average Positive Response Rate	
		Mean	SD	Mean	SD	Mean	SD
**Pre**							
**ST**	121	10.363	1.914			56.473	30.984
**ER**	122	10.000	2.263			44.262	31.620
**WH**	121	11.363	2.469			69.146	35.529
**EI**	122	9.819	1.391	13.262	1.761	84.972	21.039
**PR**	122	7.426	2.096	10.590	2.091	51.092	30.054
**DR**	119	9.025	1.820	10.050	2.629	49.859	34.953
**TF**	122	8.967	1.246			91.803	22.601
**PA**	122	7.942	1.312			70.491	32.622
**SC**	122	10.623	1.565	12.672	1.801	85.792	21.817
**Post**							
**ST**	103	11.223	2.057			71.153	29.398
**ER**	105	10.628	2.481			54.088	35.769
**WH**	105	12.647	2.556			77.358	31.722
**EI**	104	9.924	1.689	13.105	2.028	82.857	24.070
**PR**	104	8.247	2.282	9.826	2.235	42.539	32.190
**DR**	103	9.532	1.866	10.359	2.195	50.320	32.179
**TF**	104	8.798	1.585			87.142	27.759
**PA**	104	8.009	1.597			73.333	34.714
**SC**	105	10.978	1.789	12.647	2.196	79.245	27.775

Abbreviations: ST: Patient safety training received; ER: error reporting confidence; WH: working hours as an error cause; EI: error inevitability; PR: professional incompetence as an error cause; DR: disclosure responsibility; TF: team functioning; PA: patient involvement in reducing error; SC: importance of patient safety in the curriculum.

**Table 3 ijerph-19-11618-t003:** Wilcoxon signed-rank test per paired APSQ-III dimensions (pre vs. post).

Test Parameters	Dimensions								
	ST	ER	WH	EI	PR	DR	TF	PA	SC
Z score	−3.471	−2.014	−1.893	−1.026	−2.212	−0.283	−1.779	−0.790	−1.874
*p*-value	0.001	0.044	0.058	0.305	0.027	0.777	0.075	0.430	0.061

Abbreviations: ST: Patient safety training received; ER: error reporting confidence; WH: working hours as an error cause; EI: error inevitability; PR: professional incompetence as an error cause; DR: disclosure responsibility; TF: team functioning; PA: patient involvement in reducing error; SC: importance of patient safety in the curriculum.

## Data Availability

Data available on request due to privacy restrictions.

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
