# Peer review of "Knowledge and Attitudes towards Patient Safety among Students in Physical Therapy in Spain: A Longitudinal Study"

_ijerph, 2022, doi:10.3390/ijerph191811618_

Round 1
Reviewer 1 Report
Thank you for the opportunity to learn about this research and review this paper on attitudes and knowledge of students in Physical Therapy concerning Patient Safety. It is an interesting longitudinal study that applied an adapted version of the APSQ-III to 125 students before and after an internship. Although I enjoyed reading the article and recognize its value, I have some concerns that should be highlighted. (1) The first research aim does not mention that the study was developed with a particular population (i.e., students from a university in Murcia/Spain). Thus, I do not believe it should be stated as “to determine the attitudes and knowledge of students in Physical Therapy with respect to Patient Safety.” It should include “at a university in Murcia/Spain,” as the data do not fully support the generalization needed to respond to the research aim completely. I noticed that this fact is acknowledged in the study's limitations, but it is not a limitation. It is a characteristic of the research relevant enough to be stated from the beginning. Otherwise, I would say that the aim was not met. (2) The authors actually tested if their program is effective among students that underwent a particular program and a specific internship. There is not enough generalization potential, perennity, and novelty to allow this paper to be published in this journal. Please, see p.2: “to assess changes in students’ knowledge and attitudes across ‘A’ practical internship”. This means that the study assessed if that particular program/internship impacts students’ attitudes and knowledge of PS. (3) The research instrument was significantly adapted because the scales were changed. Thus, it requires the assessment of its psychometric properties. (4) “Roughly half of the students were in 4th year” (p.3). I am not sure what was covered in the curriculum up to that point, so I don’t understand the “baseline.” I should be able to understand that. Otherwise, the paper is not very informative. I cannot apply the results to other institutions; I don’t know the likely causes/influences behind the changes observed in knowledge and attitudes. (5) Considering that “a high percentage of the sample had previously received no specific content in 148 patient security (70 students; 56.0%)”, I wonder if there was nothing in the program on this topic until the 4th year. Is that correct? It does not sound like it is. Maybe only the ones that did receive it should be included in the sample (meaning that they knew the contents but had no practical experience). The ones that did not receive such content would be expected to have learned something anyway. (6) As a reader, I would have to understand in depth the internship (duration, activities, etc.) for the paper to be a meaningful reading showcasing completely understandable content.
Author Response
Please see the attachment. Thank you.
Sincerely,
The Authors

Reviewer 2 Report
It is an interesting and novel article about Knowledge and Attitudes Towards Patient Safety Among Students in Physical Therapy. Although in general it is well written and the methodology is correct, there are a few minor details I'd like to mention:
Abstract (line 18): APSQ-III. Please specify acronym.
Methods: lines 110-1. Can this change in the likert scale compromise the prior validation of the questionnaire?
The authors indicate that they recruited 125 volunteers. However, in table 2, the "n" is lower both in "Pre-test" and above all "post-test". Have the authors included in the analysis pre-test responses that did not complete the post-test survey? If so, shouldn't only those individuals who completed both surveys have been analyzed? Please clarify.
Author Response

(The authors gave the same response as above.)

Reviewer 3 Report
The article presents the changes that occur after an internship period in terms of attitudes and knowledge regarding Patient Safety of Physiotherapy students.
Some of the aspects that should be checked are the following:
-Authors: Insert a space between words on line 6: “Department of Physical Therapy, Faculty of Medicine...”
-It is suggested to write in all cases “et al” with a period at the end “et al.”.
2. Materials and Methods
-“Students were assessed of the APSQ-III questionnaire at baseline, in a first session, and then after the internship period” on line 133 . The duration of the internship for 3rd and 4th year students is very different, therefore, it is recommended to specify the duration of the internship period, and if it was the same in all cases.
-It is not clear if the initial survey was developed before imparting any content on Patient Safety.
-It is also not clear if in the periods of clinical practice the students were taught specific content on Patient Safety, or if this knowledge and attitudes are developed just by doing the practice.
3.Results
Table 1. Age put it without bold and like the rest of the variables
Table 2. If the sample is 125 students, why is the maximum N of 122 students in the pre-test? What happened to the remaining 3?
Table 3. Bold all dimensions.
4.Discussion
-It is commented that three of the authors determined that the new version of the "Attitudes to Patient Safety Questionnaire (APSQ-III): Adaptation to Physical Therapy education" adequately covers the concept measured, which confirms the apparent validity of the tool. Which authors are you referring to?
-Could the decrease in students who responded in the post-test have been controlled in some way?
-The results of the effects of clinical practices on attitudes and knowledge about patient safety in Physiotherapy students, could they be influenced by the type of practices carried out according to, for example: Physiotherapy specialty, practice center, practice tutor ...?. Some reflection on this aspect is suggested.
5.References
-Separate terms in reference nº 11: Pileta C, Robles EM, Daudinot B, et al. A tool for assessing the level of knowledge about patient's safety in undergraduate students. Revista Cubana de EducaciónMédica Superior, 2016;30:2.
-Add the DOI to the latest references.
Author Response

(The authors gave the same response as above.)
